# Genotype by environment interaction and yield stability analysis of durum wheat (*Triticum turgidum* var. *durum*) varieties using AMMI and GGE biplot analysis in Amahara Region, Northwest Ethiopia

**Alemnesh Eskezia**[1]*, **Alamir Ayenew**[2], **Yohannes Azene**[3], **Mulugeta Aytenew**[1], **Kelemu Nakachew**[1]

**1** Department of Plant Science, College of Agriculture and Natural Resources, Debre Markos University, Ethiopia, **2** Fogera National Rice Research and Training Center, Ethiopian Institute of Agricultural Research, Ethiopia, **3** Gonder Agricultural Research Center, Gondar, Ethiopia

* alemnesh_eskezia@dmu.edu.et

## Abstract

Durum wheat is a staple crop for over one- third of the global population; however, its production in Ethiopia, particularly in the Amhara Region, faces major challenges due to limited adaptable varieties, poor management practices, and environmental constraints. Newly released wheat varieties often fail in unfamiliar environments, leading to yield losses. To address this issue, a study was conducted to evaluate the adaptability, performance, and yield stability of recently released high-performing durum wheat cultivars in the Amahara region. The field trials were conducted over two consecutive years (2022/2023 and 2023/2024) at three districts; Simada, Bichena, and Dabat, using a Randomized Complete Block Design with three replications. Data were collected on main agronomic traits, including growth, phenological, and yield traits such as grain yield. Statistical analyses included SAS 9.4 and R software were used, showing significant differences in crop phenological stages, growth, and yield parameters across years and locations for tested traits. Additive Main Effect and Multiplicative Interaction(AMMI) and Genotype and Genotype × Environment (GGE) biplot analysis were used to study genotype by environmental interaction(GEI). Genotype, location, and year interactions significantly influenced all durum wheat genotypes. The combined analysis of variance for all tested traits including grain yield, showed highly significant effects (p < 0.001). Grain yield ranged from 4.5 t ha$^{-1}$ in Dabat to 2.29 t ha$^{-1}$ in Simada. Among the genotypes, Tesfaye had the highest mean yield (3.74 t ha$^{-1}$), while Wehabit had the lowest (2.75 t ha$^{-1}$). The AMMI analysis revealed that most of the variation was due to environment (45%), followed by genotype (16%) and genotype × environment interaction (16%), showing strong environmental influence on yield performance. The GGE biplot and AMMI analysis identified Tesfaye and Don metteo as stable genotypes, while Simada and

**Data availability statement:** All relevant data supporting the findings of this study are available within the manuscript and its Supporting Information files.

**Funding:** The author(s) received no specific funding for this work.

**Competing interests:** no- The authors have not declared any conflict of interest.

Bichena as representative environments. This study identifies Tesfaye as the most productive and stable wheat variety in the study area and suggests farmers should adopt it to increase wheat yield per unit area of land.

## Introduction

Durum wheat (*Triticum turgidum var. durum*) is an indigenous tetraploid wheat species (2n = 4x = 28) that has been cultivated in Ethiopia since ancient times [1–3]. The country is widely recognized as a centers of diversity for tetraploid wheat, housing a wide repertoire of landraces and genetic variation that have been cultivated for centuries [4,5]. Traditionally, this crop grown is in heavy black clay soils (vertisols) under rain-fed conditions at altitudes ranging from 1800 to 2800 meters above sea level [6]. The proportion of durum wheat in the total wheat-growing area has declined significantly due to the introduction of semi-dwarf, fertilizer-responsive bread wheat varieties, resulting in a shift where bread wheat now occupies the largest share of production [7,8]. However, the area dedicated to durum wheat is expected to increase because of rising urban demand for pasta products and the expansion of irrigated wheat production in Ethiopia [9,10].

Wheat is essential for Ethiopia's agricultural development and food security, as it is a significant source of protein and energy [11–13]. Although Ethiopia has huge potential for wheat production, the country is still unable to self sufficiency due to various production constraints. Among the constraints, the presence of environmental variation is outstanding that causes significant GEI effect for improved wheat varieties which makes them less adopted by farmers [14]. Studying GEI helps identify stable, high-yielding varieties adapted to these diverse conditions [15,16]. Wheat is crucial to the country's agricultural plans and sustainable development initiatives because of its nutritional value and versatility [17].

In Ethiopia, wheat covered approximately 1.87 million hectares, yielding approximately 5.8 million tons of grain during the 2021/22 cropping season [18]. In the Amhara region, wheat cultivation spanned 689,614.06 hectares, yielding 1.92 million tons with productivity of 2.8 t ha $^{-1}$ [18]. Overall, wheat is a staple crop in Ethiopia and is vital for food security and economic growth [19]. However, productivity remains low in many regions, including Amhara Region, dueto yield-limiting factors such as diseases, pests, climate variability, overuse of local varieties, lack of high-yielding and stable varieties, and poor communication between farmers and researchers [20]. Nowadays, Ethiopia's regional and national agricultural research institutes have released several durum wheat varieties [14,21]. Conversely, many of these varieties have not undergone comprehensive evaluation across the country, particularly in the northwestern regions [22]. To fill this gap, it is essential to generate evidence on genotypic performance and adaptability under diverse environmental conditions. Such information is crucial for devising effective intervention strategies to enhance crop productivity and promote dietary self-sufficiency [23].

Durum wheat breeding programs in Ethiopia rely on elite lines from various sources, but their performance often varies across environments due to genotype-by-environment interaction (GEI) [24,25]. To ensure stable and high-yielding varieties, multi-environment trials (METs) are essential for assessing performance and stability [26,27]. GEI complicates selection decisions because genotypes may perform inconsistently across locations, exhibiting either non-crossover (differences in magnitude) or crossover (changes in rank) interactions [28]. Therefore, relying solely on mean yield is inadequate, highlighting the need for robust GEI analysis to guide breeding and variety release [29,30].

The AMMI model integrates ANOVA for additive effects and PCA for multiplicative interactions, enabling clear stability assessment through metrics such as the AMMI Stability Value [31]. AMMI and GGE biplot were chosen due to strong GEI in the data. AMMI analyzes stability, while GGE helps visualize ideal genotypes and environments [32]. Together, they offer a comprehensive view for genotype evaluation. Although these methods have been extensively applied in crops such as like maize, rice, and bread wheat, their use in durum wheat multi-environment tr ials remains limited in Ethiopia [33,34]. Previous studies in Ethiopia and elsewhere have demonstrated the effectiveness of AMMI and GGE in identifying stable and high-yielding durum wheat varieties, such as the works of [14,32,34] and [35], which reported the identification of genotypes with broad adaptability and mega-environment delineation. Our study builds on these findings by applying both methods to durum wheat in the Amhara region under rainfed conditions, generating practical insights for stability, adaptability, and variety recommendation across diverse agro-ecologies. This dual-method approach supports robust decision-making in durum wheat breeding and variety recommendation [36,37].

Given the significant influence of environmental variability on genotype performance, stability assessment is crucial in plant breeding to identify genotypes with consistent yield across environments [38]. Stability measures, such as Shukla's stability variance, which quantifies each genotype's contribution to GEI with lower values indicating greater stability [39], and the regression-based method developed by [40],which assesses genotype sensitivity and adaptability, have been foundational in this field. Genotypic stability is commonly categorized into two forms: static stability, where genotype performance remains relatively constant across environments, and dynamic stability, where performance changes in a consistent and predictable manner in response to environmental variation [41]. These complementary concepts are critical for identifying genotypes that combine high productivity with reliable performance across diverse environmental conditions. Therefore, the current study was conducted to evaluate the adaptability, performance, and yield stability of recently released high-performing durum wheat varieties in the Amahara Region, northwestern Ethiopia.

## 2. Materials and methods

### 2.1. Descriptions of the Study area

The study was conducted across three wheat-producing districts: Simada, Bichena, and Dabat, situated within the Amhara region, Ethiopia, with permission from the Amhara Regional Agricultural Research Institute (ARARI) and Debre Markos university (DMU) (Table 1). Specifically, it was conducted at the Farmers Training Centers (FTC) in each district during the

**Table 1. Geographical description of the study area.**

| Location | CodeYear 2021/2022 | Code Year 2022/2023 | Alt. (masl) | Temp. min & max values (°C) Each year | | RF. Ave.(mm) | | Soil type | Geographic location | |
|---|---|---|---|---|---|---|---|---|---|---|
| | | | | 2022 | 2023 | 2022 | 2023 | | Lat | Lon |
| Simada | E1 | E4 | 2200 | 19-27 | 19-25 | 1012 | 1320 | Vertisol | 10 °22' N | 37° 47' E |
| Bichena | E2 | E5 | 2250 | 15.19 | 13-21 | 882 | 1256 | Vertisol | 10 38' N | 10° 37' E |
| Dabat | E3 | E6 | 2446 | 17-23 | 12-19 | 1320 | 1470 | Vertisol | 10°21` N | 37° 43`E |

Where: Alt: Altitude, Temp: temperature, and RF: Rainfall, Lat: latitude, Lon: longitude.

Source:Districta dministrative office.

main crop (rainy seasons) between June and December 2022–2024 for two consecutive years. Soils at the tested locations are Vertisols, with a pH of 5.5–7.0, moderate organic matter, and clayey texture. They contain moderate organic matter but face challenges such as phosphorus deficiency, erosion, and waterlogging in lowland areas. locations were chosen purposively based on wheat production potential and soil type.

## 2.2. Experimental material and design

Seeds of eight improved, high-yielding durum wheat varieties were obtained from the Kulumsa Agricultural Research Center, with one additional variety collected from the Adet Agricultural Research Center in Amhara, Ethiopia (Table 2). All varieties were evaluated under rainfed conditions during the 2022 and 2023 main cropping seasons and were sown on well-prepared experimental plots. These nine varieties were tested across three environments (locations). Each experimental unit (plot) consisted of nine rows with 20 cm spacing between rows, resulting in a plot area of 3.6 $m^2$ (1.8 m × 2 m). The field was plowed three times before sowing, and planting rows were prepared using a hand-pulled row marker. Seeds were manually planted using the hand drilling technique. A spacing of 0.5 m was maintained between plots, and 1 m was left between blocks. Since the two years (2022 and 2023) were also considered as blocks, the six combinations of the field blocks and years were used as blocks in the multilocation trial model.

For each plot, a standard fertilizer application of 100 kg/ha NPS and 150 kg/ha urea was applied following durum wheat recommendations. Half of the urea and the full amount of NPS were applied during planting, while the remaining half of the urea was applied at the mid-tillering stage. A seed rate of 150 kg/ha was used. Weeding was performed manually across all experimental fields at the three locations to manage weeds. No herbicides or fungicides were used for weed or disease control.

## 2.3. Data collected

The data for the eleven traits were collected on a plant-and-plot basis. Data were obtained on a plot basis from all plants in the central rows of each plot, leaving two border row from each side of the plots. Ten representative plants per plot were

Table 2. Variety used in the study.

| S. N | Variety code | Variety name | Pedigree | Year of release | Center | Matu rity period | Disease resistance levels | Yied on station | Area of daptation |
|---|---|---|---|---|---|---|---|---|---|
| 1 | G1 | Fetan | CDSS02B00643 S-0Y-0M-1Y4M-04-Y-0B-2Y | 2018 | DZARC | 95-110 | Moderately resistance | 46-50 | 500-2600 |
| 2 | G2 | Don Matteo | 43IDSNMeh 2011−82 | 2018 | CGS Italian | 115-130 | Moderately resistance | 50-55 | 1500-2750 |
| 3 | G3 | Alem-tena | 43IDSNMeh 2011−130 | 2015 | DZARC | 95-110 | Moderately resistance | 44-50 | 500-2600 |
| 4 | G4 | Tesfaye | ARMENT//SRN-NIGRIS-4/3/CANED-9.1/4/ TOSKA-26RASCON-37//SNITSN/5/PLAYERO | 2017 | DZARC | 110-130 | Moderately resistance | 50-55 | 1600-2800 |
| 5 | G5 | Weha-bit | Waha (plc/Ruff//Gta/Rtte) | 2017 | Mekelle University | NA | Moderately resistance | NA | NA |
| 6 | G6 | Rigeat | Acc#208304 | 2017 | Mekelle University | NA | Moderately resistance | NA | NA |
| 7 | G7 | Utuba | (= Icajihan42) Omruf1/Stojocri2/3/1718/ BeadWheat24//Karimï¿½ | 2015 | DZARC | 105-118 | Moderately resistance | 53-57 | 1880-2700 |
| 8 | G8 | Man-gudo | Mangudo/Mekuye/DZ 2013 meh DW F1 P#20/DZ 2014 meh DW F2 P#19−8 | 2012 | SARC/ORARI | 110-124 | Moderately resistance | 45-50 | 1880-2700 |
| 9 | G9 | Toltu | 4/B/R9096#21001(980SN Patho) | 2010 | SARC/ORARI | NA | Moderately resistance | NA | NA |

randomly selected from the central rows for recording data at the plant level. Agronomic traits, including crop phenology, growth, yield, and yield components were assessed during the study period. Crop phenology included maturity day and days to heading, while growth parameters, such as plant height, number of productive tillers, number of grains per spike, number of spikelets per spike, spike length, and yield-related traits, including biomass yield, thousand seed weight, number of seed per spikelet, grain yield, and harvest index were recorded.

**Days to heading (days):** The number of days from sowing until the tips of the spike first emerged from the main shoot in 50% of the plant population within a plot was recorded.

**Days to 90% maturity (days):** The number of days from sowing to physiological maturity was recorded. Physiological maturity was determined when 90% of the plant in a plot showed a light yellow (straw) color in their stems, leaves, and floral bracts.

**Number of productive tillers (no):** The number of productive tillers was determined at maturity by counting all spike-producing tillers on 10 plants per plot.

**Number of spikelets per spike (no):** The number of spikelets per spike was counted from ten representative spikes per plant, and the average was calculated.

**Plant height (cm):** Plant height was measured from the base of the main stem to the tip of the spike using an average of ten randomly selected plants in the central four rows of each plot.

**Spike length (cm):** Spike length was recorded at the physiological maturity stage by measuring the middle rows of 10 randomly tagged plants from the base of the tip of the spike (excluding the awns), and the average length was calculated.

**Biomass yield (ton ha$^{-1}$):** Aboveground total biomass was recorded after sun-drying, attained constant weight, and then converted to t ha$^{-1}$.

**Grain yield (ton ha$^{-1}$):** The weight of air-dried seeds harvested from each plot was recorded, and the yield from the net plot area was converted into t ha$^{-1}$

**Harvest index (%):** Harvest index was calculated as the ratio of grain yield per plot to the biological yield per plot, expressed as a percentage.

$$HI\ (\%)\ =\ \frac{Grain\ Yield}{Biomass\ Yield}\ x\ 100$$

## 2.4. Data analysis

Data were analyzed using R statistical software version 4.4.1 (R Core Team, 2021). Analysis of variance (ANOVA) was conducted on the multi-environment data using the aov function from the car package [42] to evaluate the significance of genotype-by-environment interactions. The GGE model function from the GGE Biplots package was applied to the genotype-by-environment data matrix [43]. Subsequently, the "Which-Won" function was used to generate the "which-won-where" view of the GGE biplot. The Env relationship function in the same package was utilized to plot environmental relationships, the Disc Rep function was used to assess discrimination ability and representativeness, and the Mean Stability function was applied to create the discrimination ability and representativeness view and stability plots.

AMMI analysis was carried out using the AMMI function in the agricolae package [44].Genotype stability was further assessed using additional stability parameters, including the coefficient of variation (CV) [45] Shukla's stability variance [39], AMMI Stability Value (ASV) [46], ecovalence (Ecoval) [47], and deviation from joint regression analysis (Sij) [48]. These parameters were estimated using the metan package in R [49].

The ASV measures the stability of genotypes based on their interactions with the environment. Lower ASV values indicate more stable genotypes.

AMMI stability value was calculated as follows:

$$ASV = \sqrt{[(SSIPCA1\ /\ SSIPCA2)\ (IPCA1\ score)]2\ +\ (IPCA2\ score)2}$$

Where; SSIPCA1 and SSIPCA2 are the sums of the squares of the first and second principal component axes (IPCA), respectively. The IPCA 1 and IPCA2 scores are the interaction scores for the first and second IPCAs, respectively.

GGE biplot analysis is a graphical method that evaluates genotype performance and stability across environments by combining genotype and genotype-environment interaction effects, helping to identify the best and most adaptable genotypes.

$$Y_{ij} - \mu - \beta_j = \sum_{k=1}^{n} \lambda_k \xi_{ik} \eta_{jk} + \varepsilon_{ij}$$

Where:

- $Y_{ij}$ is the grain yield of genotype i in environment j,

- $\mu$ is the grand mean,

- $\beta_j$ is the main effect of environment j,

- $\lambda_k$ is the singular value for principal component k,

- $\xi_{ik}$ and $\eta_{jk}$ are the genotype and environment scores for PCk, respectively,

- $\varepsilon_{ij}$ is the residual.

## 2.6. Ethics approval and consent to participate

- This study involved plant materials only and did not include human participants or experimental animals.

## 3. Results and discussion

### 3.1. Combined analysis of variance (ANOVA)

The analysis of variance revealed highly significant differences ($P < 0.01$) among varieties for the traits examined (Table 3). Locations also showed highly significant difference ($p < 0.01$) for days to heading, tiller number, spike length, number of seeds per spikelets, plant height, thousand seed weight, biomass yield, grain yield, and harvest index. Additionally, varieties demonstrated highly significant variation ($P < 0.01$) among all evaluated traits. The year showed a highly significant effect ($p < 0.01$) on plant height, spike length, biomass yield, grain yield, and thousand seed weight.

The interaction between variety and location (V x L) was highly significant ($P < 0.01$) for all traits tested in the durum wheat varieties. Similarly, the interaction between variety and year (V x Y) was highly significant ($p < 0.01$) for plant height, number of grains per spike, grain yield, and harvest index. The interaction between location and year (L x Y) was also highly significant ($p < 0.01$) for all tested traits except for the number of seeds per spikelet and biomass yield.

The three-way interaction effect of variety, location, and year (V x L x Y) was highly significant ($p < 0.01$) for days to heading, plant height, tiller number, spike length, number of seeds per spikelets, thousand seed weight, biomass yield, grain yield, and harvest index. The main effect of location by year (LxY) had highly significant ($p < 0.01$) effect on days to heading, maturity day, plant height, and grain yield. However, non significant effect was observed for the number of seeds per spikelet and biomass yield. The interaction effect of location by year had a highly significant ($P < 0.01$) effect on days to heading, maturity day, plant height, and grain yield, however, non significant effect was observed for the number of seeds per spikelet and biomass yield. As shown in Table 3, the interaction effect of location and variety had a significant ($P < 0.05$) effect on all evaluated traits. The interaction effect of variety by year was highly significant ($P < 0.01$) for plant

**Table 3. Mean squares ofthe two years (cropping season) combined ANOVA results of Eleven quantitative traits from three locations.**

| SOV | Mean squares | | | | | | | | | |
|---|---|---|---|---|---|---|---|---|---|---|
| | DH | DM | PH | TN | SL | NSPS | TSW | BY | GY | HI |
| Genotypes (G) | 345.28** | 789.84** | 585.36** | 1.36** | 11.475** | 34.85** | 226.28** | 5.99** | 2.07** | 86.20** |
| Location (L) | 6.968ns | 194.93ns | 41.61** | 47.51** | 43.41** | 24.83** | 234.07** | 25.21** | 4.319** | 165.47** |
| Year (Y) | 0.964ns | 89.92ns | 164.36** | 0.03ns | 0.30ns | 2.45ns | 27.16** | 3.43** | 4.200** | 1.05ns |
| V x L | 42.69* | 187.61* | 222.18** | 2.02** | 11.54** | 23.51** | 120.06** | 2.49** | 0.511** | 54.12** |
| V x Y | 39.36ns | 95.88ns | 37.53** | 0.59** | 1.29** | 7.30** | 22.20** | 0.42ns | 0.98** | 13.19ns |
| L x Y | 289.05** | 496.186* | 173.19** | 1.49** | 2.57** | 3.25ns | 95.82** | 0.750ns | 2.003** | 59.46** |
| L x rep | 33.73ns | 188.16ns | 6.47ns | 0.22ns | 1.36** | 5.10ns | 12.34** | 0.63ns | 0.037ns | 27.26ns |
| V x L x Y | 53.39** | 152.547ns | 33.32** | 0.33** | 1.35** | 3.43ns | 18.86** | 1.015** | 0.859** | 53.28** |
| Residuals | 22.59 | 103.58 | 10.45 | 0.12 | 0.33 | 2.74 | 2.53 | 0.41 | 0.02 | 14.35 |

Where SOV: source of variation; **: highly significant at $P < 0.01$; *: significant at $P < 0.05$; ns: non-significant; DF: Degrees of freedom; DH: days to heading; DM: days to maturity; PH: plant height; TN: tiller number; SL: Spike length; NSPS: number of spikelets per spike; BY: biomass yield; NSS: number of seed per spike, TSW: thawed seed weight, GY: grain yield; and HI: harvest index.

height, number of seeds per spikelet, biomass yield, grain yield, and spike length. Conversely, a non-significant effect was observed for maturity date and days to heading.

Overall, the combined analysis revealed significant variations among the nine durum wheat varieties for most traits. These findings are consistent with previous studies by [50], which reported significant differences (p < 0.05) among durum wheat genotypes across all individual test locations over multiple years.

Days to heading, plant height, tiller number, biomass yield, grain yield, and harvest index were all highly significant (p ≤ 0.01), as shown in Table 3. However, maturity day and the number of seeds per spike were not influenced by the interaction effects of year, variety, and location. In line with the present results, previous studies have reported highly significant differences ($p ≤ 0.01$) [51]. Significant genotype-by- location interactions, with highly significant differences (p ≤ 0.01) among food barely varieties for days to maturity and grain yield was also report by [52]. The observed significant GEI interaction indicate inconsistent performance across the tested environments and highlight the varying discriminative abilities of these environments. This suggests the possibility to identify high-yielding and stable genotypes. A particular crop variety may perform well in one environment but poorly in another due to variations in climate, soil conditions, or management practices [53,54].

The results of this study are consistent with previous research by [55], which reported significant difference (P < 0.05) among bread wheat genotypes for grain yield at all individual test locations in multi-environment trials.

### 3.2. Performance of the genotype

The grain yield performance of the durum wheat varieties showed significant variability across the tested locations. Tesfaye (G4) achieved the highest grain yield 3.74 t ha⁻¹, indicating its superior adaptability and productivity under the test conditions. It was followed by Don Matteo (G2) which achieved a yield of 3.62 t ha⁻¹, also demonstating strong performance. In contrast, Wehabit (G5) and Rigeat (G6) exhibited the lowest grain yields, at 2.75 t ha⁻¹ and 2.78 t ha⁻¹, respectively, suggesting their limited potential for high productivity in these environments.

The differences in grain yield among the genotypes can be attributed to variations in key yield components such as tiller number, spike length, and thousand seed weight. These findings underscore the importance of selecting high-yielding genotypes, such as Tesfaye (G4) to enhance durum wheat production in the study area. Our results are consistent with those of [56] and [57], who found that durum wheat genotypes varied significantly in terms of adaptation and grain yield across Ethiopian highlands. Tesfaye and Don Matteo, in particular were found to be promising genotypes that consistently performed well

in a variety of environmental settings. Similar to the current study, [58] observed that cultivars like Wehabit showed reduced grain yield stability. While Wehabit and Rigeat's subpar performance highlights the necessity for location-specific recommendations, Tesfaye's improved performance across locales demonstrates its appropriateness for wider adaption.

The observed variability in grain yield reflects the influenced GEI interaction. High-yielding genotype, such as Tesfaye (G4) 3.74 t ha$^{-1}$ demonstrate better adaptability and efficient resource utilization, whereas low-yielding genotype like Wehabit (G5) 2.75 t ha$^{-1}$ show poor adaptability to the tested environments (Table 4). This result further supports the importance of evaluating durum wheat genotypes across multiple environments to identify stable and high-yielding varieties. Selecting adaptable varieties like Tesfaye (G4) will be vital to enhance durum wheat production and support food security goals in Ethiopia.

### 3.3. Genotype performance at each tested location

The mean grain yield of the nine durum wheat genotypes across six environments (E1–E6) showed notable GEI, highlighting differential performance under varying conditions (Table 5). Tesfaye (G4) consistently exhibited the highest mean grain yield (3.74 t ha$^{-1}$), driven by exceptional adaptability in environments E1 (3.97 t ha$^{-1}$), E4 (3.97 t ha$^{-1}$), and E6 (4.50 t ha$^{-1}$). This indicates that Tesfaye is particularly well-suited to a range of environmental conditions. Don Matteo (G2) ranked second with (3.62 t ha$^{-1}$), excelling in E2 (4.70 t ha$^{-1}$) and E4 (4.13 t ha$^{-1}$), suggesting suitability for high-yielding environments, particularly in Simada and Bichena. In contrast, Wehabit (G5) and Rigeat (G6) recorded the lowest mean grain yield, at 2.75 t ha$^{-1}$ and 2.78 t ha$^{-1}$ respectively. Their consistently poor performance across most environments suggests limited adaptability or a higher sensitivity to environmental variations, making them less reliable for achieving high yields. Similarly, Alemtena (G3), despite excelling in E3 (4.82 t ha$^{-1}$), had a lower overall mean yield (3.08 t ha$^{-1}$) due to inconsistent performance in other environments, indicating that its high yield potential is not consistent across all conditions. Previous studies in Ethiopia support these findings. For instance, Tesfaye and Don Matteo were identified as stable and high-yielding durum wheat varieties across several northern Ethiopian locations [25,59]. Similarly, [60] reported that Tesfaye demonstrated broad adaptability and consistent yield performance in both highland and moisture-stressed areas. Conversely, [60] found that Wehabit exhibited poor stability across diverse environments, aligning with our results on its limited performance.

Table 4. Mean grain yield and yield-related traits of durum wheat genotypes tested at three locations for two years.

| Genotypes | DH | DM | PH | TN | SL | NSPS | TSW | BY | GY | HI |
|---|---|---|---|---|---|---|---|---|---|---|
| 1 | 62.00[b] | 104.44[bcd] | 87.17 cd | 3.81 cd | 7.62[d] | 19.43[a] | 35.47[c] | 10.47[b] | 3.3[c] | 26.74[e] |
| 2 | 61.22[b] | 110.72[ab] | 98.01[a] | 4.01[abc] | 9.83[a] | 19.39[a] | 36.99[b] | 11.08[a] | 3.62[b] | 30.81[abc] |
| 3 | 53.44[c] | 93.33[f] | 88.29[c] | 3.45[e] | 7.69[d] | 16.36[c] | 34.46 cd | 9.64[c] | 3.09[d] | 31.25[ab] |
| 4 | 70.72[a] | 113.39[a] | 94.23[b] | 4.18[a] | 9.06[b] | 19.36[a] | 41.23[a] | 10.94[a] | 3.74[a] | 33.17[a] |
| 5 | 62.11[b] | 106.13[bc] | 83.55[e] | 4.11[ab] | 8.47[c] | 19.16[a] | 31.44[f] | 9.40[c] | 2.75[f] | 28.37[cde] |
| 6 | 61.44[b] | 101.78[cde] | 78.53[f] | 3.9[bc] | 8.1[c] | 16.64[bc] | 31.12[f] | 9.64[c] | 2.78[f] | 27.77[de] |
| 7 | 63.39[b] | 95.23[ef] | 87.02 cd | 3.55[e] | 7.47[d] | 16.42[c] | 29.5[g] | 10.08[b] | 2.95[e] | 28.76[bcde] |
| 8 | 62.89[b] | 102.61 cd | 85.80[d] | 4.19[a] | 8.29[c] | 17.66[b] | 33.68[de] | 10.15[b] | 3.12[d] | 29.65[bcd] |
| 9 | 63.06[b] | 98.97[def] | 85.36[de] | 3.66[de] | 7.5[d] | 16.89[bc] | 32.63[e] | 10.18[b] | 3.15[d] | 26.63[e] |
| MSE | 22.59 | 103.58 | 10.45 | 0.12 | 0.33 | 2.74 | 2.53 | 0.41 | 0.02 | 14.35 |
| Mean | 62.25 | 102.96 | 87.55 | 3.87 | 8.22 | 17.92 | 34.06 | 10.18 | 3.17 | 29.24 |
| CV | 7.64 | 9.88 | 3.69 | 9.04 | 7.03 | 9.23 | 4.67 | 6.27 | 4.88 | 12.95 |
| LSD | 1.81 | 3.88 | 1.23 | 0.13 | 0.22 | 0.63 | 0.61 | 0.24 | 0.06 | 1.45 |

Where: MSE = Mean sequar of error LSD = List Significant Deference and CV = Cofficent of varation.

**Table 5. Mean grain yield (t ha⁻¹) of nine durum wheat genotypes across six environments (location and year combinations).**

| Genotypes | (E1) | (E4) | (E3) | (E6) | (E2) | (E5) | Mean |
|---|---|---|---|---|---|---|---|
| 1 | 3.53b | 3.66b | 2.57d | 3.67b | 3.13a | 3.2b | 3.29 |
| 2 | 3.84a | 4.13a | 2.6d | 4.7a | 3.17a | 3.26ab | 3.62 |
| 3 | 2.74c | 2.91c | 4.82a | 2.82c | 2.41 cd | 2.8c | 3.08 |
| 4 | 3.97a | 3.97a | 3.23b | 4.5a | 3.23a | 3.53a | 3.74 |
| 5 | 2.87c | 2.79 cd | 2.75 cd | 2.83c | 2.5bc | 2.77c | 2.75 |
| 6 | 2.7c | 2.63d | 2.92c | 3.63b | 2.29d | 2.53c | 2.78 |
| 7 | 2.85c | 2.85 cd | 3.18b | 3.41b | 2.61b | 2.8c | 2.95 |
| 8 | 3.97a | 2.91c | 2.92c | 3.63b | 2.5bc | 2.77c | 3.12 |
| 9 | 2.85c | 2.85 cd | 2.57d | 4.7a | 2.41 cd | 3.53a | 3.15 |
| MSE | 0.03 | 0.023 | 0.015 | 0.031 | 0.007 | 0.03 | |
| Mean | 3.256 | 3.191 | 3.062 | 3.767 | 2.695 | 3.02 | 3.16 |
| CV | 5.314 | 4.773 | 4.035 | 4.642 | 3.125 | 5.62 | |
| LSD | 0.299 | 0.264 | 0.214 | 0.303 | 0.146 | 0.29 | |

Where: E1 = Bichena 2022, E2 = Simada 2022, E3 = Dabat 2022 = E4 = Bichena 2023 = E5 = Simada 2023, E6 = Dabat 2023, MSE = Mean sequar of error and CV = Cofficent of varation.

Environmental effects were evident, with Dabat 2023 (E6) producing the highest mean grain yield (3.767 t ha⁻¹), followed by Bichena 2022 (E1) and Bichena 2023 (E4), which recorded mean grain yielda of 3.256 t ha ⁻¹and 3.191 t ha⁻¹, respectively. Conversely, Simada 2022 (E2) had the lowest mean yield (2.695 t ha⁻¹), likely due to less favorable growing conditions in that region. These results highlight the importance for evaluating genotypes across diverse environments to identify high-yielding and stable performers. Genotypes such as Tesfaye (G4) and Don Matteo (G2) should be prioritized for large-scale production in the Amhara region and similar agro-ecologies, given their demonstrated adaptability and productivity. These findings contribute to the growing body of evidence supporting the use of multi-environment trials and stability analysis to guide durum wheat variety recommendations in Ethiopia.

### 3.4. Additive main effects and multiplicative interaction (AMMI) analysis

Additive Main effect and Multiplicative Interaction (AMMI) analysis of variance for grain yield of the nine durum wheat genotypes evaluated across six environments is presented in (Table 6). The AMMI analysis revealed genetic variation and potential for selecting stable genotypes. The results showed that variation due to E, G, and GEI were highly significant ($P<0.01$). The sum squares (SS) partitioning indicated that the environmental effect was the largest contributor to variation, followed by GEI and genotype main effect. Since changes in G and G×E are ordinarily smaller, the environment often explains most of the differences in genotype performance [61]. Furthermore, the AMMI model's first two principal component axes (IPCAs) were found to be highly significant ($P<0.01$) when used for GEI partitioning. The AMMI, which includes IPCA1 and IPCA2, is the best prediction model for cross-validating yield variation explained by the genotype-environment interaction [62,63]. Variations in the grain yield components from the AMMI indicated that genotype, environment, and genotype by environment interaction (GEI) all had significant ($P<0.01$) effects, explaining variability in both environments and genotypes, and that there are possibilities for the selection of genotypes that are high-yielding, stable, and well-performing. G8 (Mangudo) was the most stable but only modestly productive, according to the AMMI1 biplot, while G4 (Tesfaye) and G2 (Don Matteo) combined high yield with good stability. Although they demonstrated strong interactions with particular surroundings, G3 (Alemtena) and G6 (Rigeat) were less stable. In contrast to E3 (Dabat 2022), which was highly interactive and showed significant environmental influence, E6 (Dabat 2023) was both stable and high yielding.

**Table 6. AMMI analysis of variance for grain yield of nine durum wheat genotypes.**

| Source | DF | Sum square | Mean square | Proportion | Accumulated |
|---|---|---|---|---|---|
| ENV | 5 | 29.78 | 3.36** | 45.22% | NA |
| REP(ENV) | 12 | 0.49 | 0.04** | 0.74% | NA |
| GEN | 9 | 16.6 | 2.08** | 25.2% | NA |
| GEN: ENV | 40 | 16.84 | 0.75** | 25.86% | NA |
| PC1 | 12 | 20.72 | 1.72** | 69.6 | 69.6 |
| PC2 | 10 | 5.62 | 0.56** | 18.9 | 88.5 |
| PC3 | 8 | 2.11 | 0.26** | 7.1 | 95.6 |
| PC4 | 6 | 1.11 | 0.19** | 3.7 | 99.3 |
| PC5 | 4 | 0.22 | 0.06** | 0.7 | 100 |
| Residuals | 96 | 2.15 | 0.02 | NA | NA |
| Total | 201 | 65.86 | 0.47 | NA | NA |

Genotype rank based on AMMI stability value

| Number | Genotype | Mean yield | Rank | ASTAB | ASTAB | Shukula | bij | ASI | IPCA 1 | IPCA 2 | ASV | Rank |
|---|---|---|---|---|---|---|---|---|---|---|---|---|
| 1 | Fetan | 3.295 | 3 | 3.89 | 5 | 1.23 | 8.36 | 2.76 | −0.415 | 0.2910 | 1.38 | 7 |
| 2 | Don Matteo | 3.51 | 2 | 5.78 | 6 | 2.22 | 1.41 | 4.00 | −0.609 | 0.2569 | 1.99 | 8 |
| 3 | Alemtena | 3.01 | 6 | 1.43 | 9 | 7.42 | −1.65 | 7.56 | 10.161 | 0.1107 | 3.77 | 9 |
| 4 | Tesfaye | 3.69 | 1 | 1.41 | 1 | 2.49 | 1.20 | 1.74 | −0.264 | 0.1547 | 8.73 | 5 |
| 5 | Wehabit | 2.75 | 9 | 2.93 | 3 | 3.37 | 3.46 | 1.07 | 0.1521 | 0.2138 | 5.38 | 2 |
| 6 | Rigeat | 2.78 | 8 | 3.64 | 4 | 5.04 | 1.31 | 1.11 | 0.1366 | 0.3365 | 5.57 | 3 |
| 7 | Utuba | 2.94 | 7 | 1.76 | 2 | 2.60 | 7.32 | 1.74 | 0.2651 | −0.1175 | 8.69 | 4 |
| 8 | Mangudo | 3.11 | 4 | 7.53 | 7 | 1.50 | 1.42 | 9.67 | 0.1166 | 0.2995 | 4.83 | 1 |
| 9 | Toltu | 3.07 | 5 | 9.41 | 8 | 2.74 | 1.73 | 2.66 | −0.3087 | −0.8727 | 1.32 | 6 |

Grain yield was significantly impacted by environments, genotypes, and their interactions, according to AMMI analysis (P < 0.001). The sum of squares resulting from the GEI is further separated into five important Interaction Principal Components (PC-1 to PC-5) in consideration of the multiplicative component of AMMI (Table 6). The highest proportion of the total sum of squares was accounted for by the environment (45.96%), followed by the genotype by environment interaction (25.86%) and then genotype (25.2) (Table 6). According to Gauch (2006), this is consistent with the outcome anticipated from GEI analysis. The environment-related sum of squares' maximum magnitude indicates that the test conditions vary and have an impact on how the grain yield characteristic manifests itself. In line with findings from barley [64] and durum wheat grain yield [65] the GEI and the largest percentage of environmental variance are in accord. The higher sum of squares due to the genotype-by-environment interaction compared to the genotype effect indicates the presence of mega-environments (Fig. 1). Based on IPCA1 and IPCA2, Tesfaye, Fetan, and Don Matteo showed small interaction values and registered high grain yield, indicating wide adaptability and stability, while Alemtena had large IPCA scores, reflecting poor stability. Mangudo and Rigeat showed moderate adaptability. The findings reveal that the various genotypes reacted differently to the environmental conditions, highlighting the phenotypic diversity among the evaluated genotypes. This also implies the potential presence of distinct mega-environments, each with genotypes that exhibit varying performance levels.

## 3.5. Stability analysis and mega-environment classification using GGE biplot

### 3.5.1. GEI analysis for which-won-where patterns using GGE biplot.
The GGE results were similar to those obtained from AMMI and combined ANOVA. The polygon view of the GGE biplot highlights the top-performing

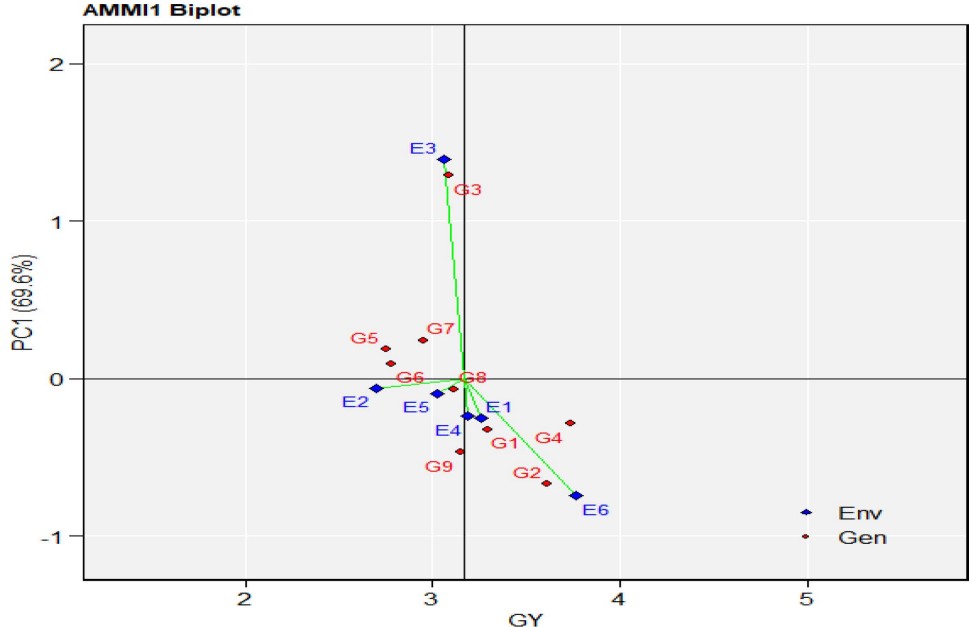

**Fig 1. AMMI Analysis model biplot for grain yield of durum wheat genotypes evaluated at six environments.** Where: G1: Fetan,G2: Don Matteo, G3: Alemtena, G4: Tesfaye, G5: Wehabit, G6: Rigeat, G7: Utuba, G8:Mangudo, and G9: Toltu genotype (E1:Simada, E2: Bichena, E3: Dabat 2022 crop-ping season and E4: Simada, E5: Bichena, E6: Dabat in 2023 cropping season. The red numbers connected to the polygon n represent the genotypes, whereas the blue numbers represent the locations/test environments. Simada, E2: Bichena, E3: Dabat 2022 cropping season and E4: Simada, E5: Bichena, E6: Dabat in 2023 cropping season. The red numbers connected to the polygon represent the genotypes, whereas the blue numbers represent the locations/test environments.

genotype in each environment, visually representing key concepts such as crossover interaction, mega-environment differentiation, and specific adaptation [66]. Mega environment analysis refers to dividing a crop-growing region into distinct target zones [67].

In this analysis, six vertex genotypes were identified: G3 (Alemtena), G4 (Tesfaye), G2 (Don Matteo), G5 (Wehabit), G6 (Rigeat), and G9 (Toltu). The biplot was divided into six sections by using six rays, with environments distributed across three of these sections while genotypes appeared in all. Each sector's vertex genotype exhibited the highest grain yield for the environments within that sector, indicating specific adaptation to particular mega-environments.

The GGE biplot analysis identified two distinct wheat-growing mega-environments.The first mega-environment com-prised environments E1 (Bichena 2022), E2 (Simada 2022), E4 (Bichena 2023), E5 (Simada 2023), and E6 (Dabat 2023), with G2 identified as the winning genotype. The second mega-environment consisted solely of E3 (Dabat 2022), where G3 was the top-performing genotype. No mega-environments were associated with sectors where genotypes G5, G6, and G9 were vertex genotypes, indicating that these genotypes were not the best performers in any of the tested environment. G8, located near the origin, exhibited average performance with wide adaptability but moderat grain yield. The GGE biplot explained 82.67% of the total G × E variation (Axis 1: 61.48%, Axis 2: 21.19%). Previous studies by [68] have reported similar findings, suggesting that an ideal genotype should have high mean performance and stability. The genotypes that performed the best or worst in specific environments were the furthest from the biplot's origin [69,70], which were more sensitive to environmental changes, and regarded as specifically adapted genotypes. The GGE biplot polygon view helps identify the most ideal test environments within each sector, which are crucial for selecting superior genotypes [71]. Therefore, G4 (Tesfaye) had the highest yield in Bichena and Simada 2022, G9 (Toltu) had the highest yield in Simada and Dabat, and G3 (Alemtena) had the highest yield in Dabat 2022. The most effective method for identifying superior

genotypes and visualizing the patterns of interaction between genotypes and environments in data analysis is the polygon view of the GGE biplot, which helps to estimate the potential existence of various mega-environments [72,73]. Vertex genotype G4 falls in the first and second quadrants, G9 falls in the third quadrant whereas G3 is located in the second and fourth quadrants. Therefore, the six testing environments were categorized into three mega environments, while the nine genotypes were divided into six genotypic groups (Fig 2).

**3.5.2. Evaluation of environments relative to the ideal environment.** GGE biplot representation from the center of the concentric circles in a polyg on perspective, the discrimination and representativeness of the GGE biplot often illustrate the ranking of an ideal test environment for a particular genotype or for all genotypes (Fig 3). A point located on the average environment axis in the positive direction, with the longest vector from the biplot origin is considered the most representative and thus the optimal test environment in the GGE biplot [66,74]. The test environment that is both most representational of the target environment and most discriminating (informative) is deemed optimal. The ideal test environment is represented by the center of the concentric circles (Fig 3). E5 and E2 performed the worst in selecting genotypes adapted to the entire region, whereas E1, E4, and E6 were closest to the ideal environment point and, therefore, best represented the Durum Wheat production environments.

**3.5.3. Evaluation of genotypes relative to the ideal genotype.** The ideal genotype for durum wheat production is positioned at the center of the concentric circles, in the GGE biplot symbolizing optimal traits characterized by high mean yield and stability. In this context, Tesfaye, stands out desirable genotype due to its exceptional yield performance and stable characteristics, making it highly suitable for grain production. Genotypes that are close to the ideal genotype, such

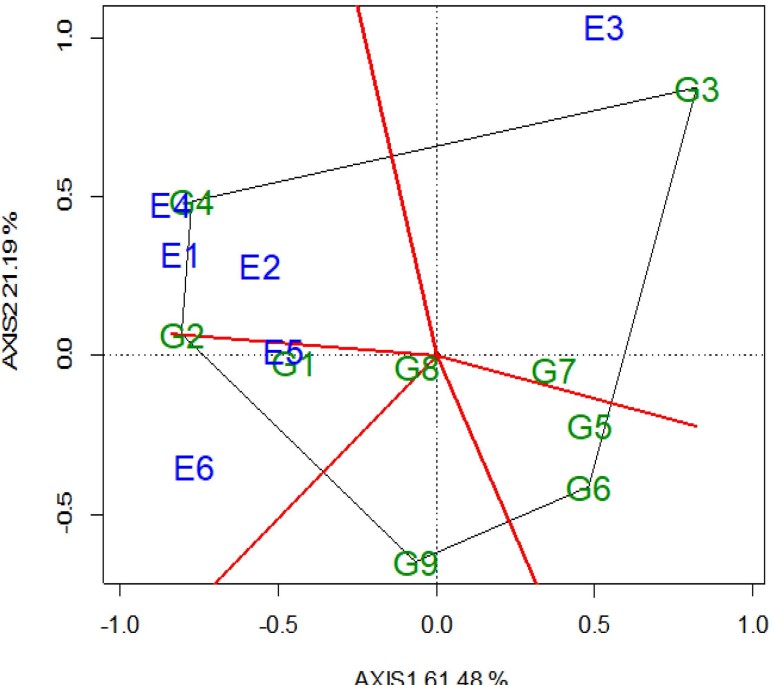

**Fig 2. Which-won-where view of GGE biplot for grain yield of durum wheat genotype.** Where: G1: Fetan, G2: Don Matteo, G3: Alemtena, G4: Tesfaye, G5: Wehabit, G6: Rigeat, G7: Utuba, G8:Mangudo, and G9: Toltu genotype (E1: Simada, E2: Bichena, E3: Dabat 2022 cropping season and E4: Simada, E5: Bichena, E6: Dabat in 2023 cropping season. The blue numbers connected to the polygon represent the genotypes, whereas the green numbers represent the locations/test environments.

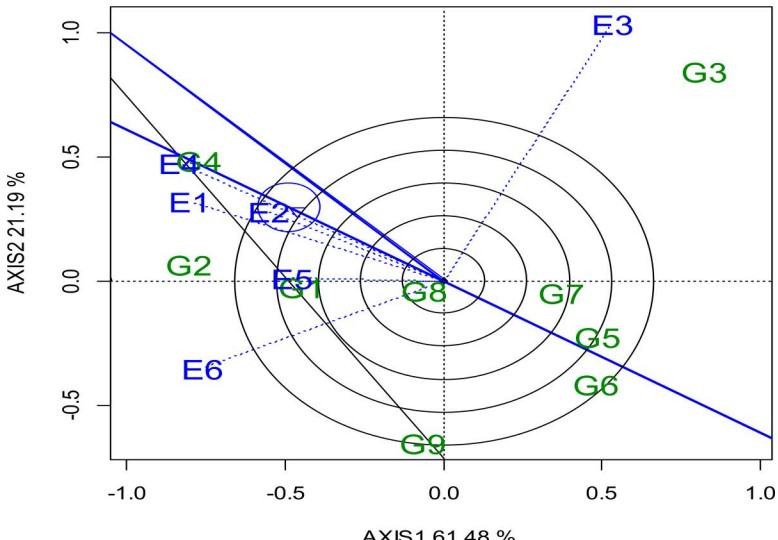

**Fig 3. Discrimination vs. representativeness of genotypes.** Where: E1: Simada, E2: Bichena, E3: Dabat 2022 cropping season and E4: Simada, E5: Bichena, E6: Dabat in 2023 cropping season.

as Fetan (G1), Utuba (G7), and Wehabit (G5), are also recognized for their favorable performance and stability, aligning closely with the traits of the optimal genotype. These genotypes are therefore considered strong candidates for cultivation.

In contrast, genotypes positioned further from the ideal center specifically Don Matteo (G2), Toltu (G9), and Alemtena (G3) are considered less desirable due to their low yield performance (Fig 4). G8 (Mangudo) was the most stable but only modestly productive, according to the grain yield men values and stability figure. This classification is supported by data from the mean separation test referenced in Fig 4, which corroborates these observations regarding yield stability and performance. Overall, the analysis highlights the value of selecting genotypes based on their proximity to the ideal genotype as depicted in the concentric circle model, of GGE biplot. This approach effectively illustrates the relationship between yield performance and genetic stability in durum wheat breeding programs. Supporting this finding, [75] identified variety G-169 as the most ideal genotype for durum wheat grain production. G4 as the ideal genotype and E4 as the ideal environment. G1 and G2 also appear as desirable genotypes, while E1 and E2 emerge as desirable environments, suggesting they offer favorable conditions for good performance and stability (Fig 4). Additionally, Multivariate analyses of Ethiopian durum wheat revealed genotypes that are both high-yielding and stable across trials [25]. The relative contributions of stability and grain yield for identifying desirable genotypes found in this study, using the ideal genotype procedure of the GGE biplot, were also similar to those reported by Adem Wado [76] genotype × location × planting pattern interaction and stability analysis of sorghum (Sorghum bicolor (L.) Moench) varieties in the drylands of Ethiopia.

**3.5.4. Interrelationship among environment.** In a GGE biplot, the correlation intensity between enviroment is represented by the cosine of the angle between any two environment vectors [66]. A correlation is considered positive if the angle is less than 90°, negative if it is greater than 90°, and nonexistent if it is around 90°. According to [77] and [61] there is a significant connection between the correlation coefficient of two environments and the angle between their vectors, which is utilized to categorize the test environments. Therefore, the top-yielding genotypes in one environment will perform best in other environments if two environments have a positive correlation. In the context of GGE biplot analysis,understanding the relationship between environments is crucial for interpreting genotype performance across different conditions. When two environments show a negative correlation, it suggests that genotypes performing well

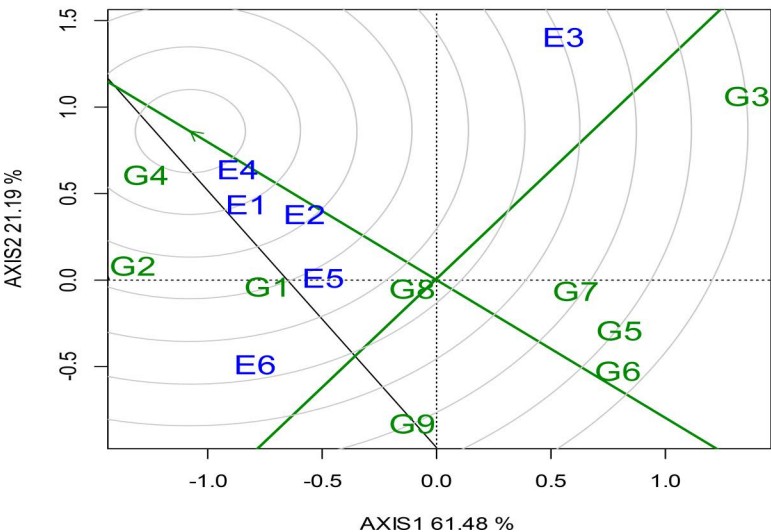

**Fig 4. Ranking of genotypes based on yield performance and stability.** Where: G1: Fetan, G2: Don Matteo, G3: Alemtena, G4: Tesfaye, G5: Wehabit, G6: Rigeat, G7: Utuba, G8:Mangudo, and G9: Toltu genotype (E1: Simada, E2: Bichena, E3: Dabat 2022 cropping season and E4: Simada, E5: Bichena, E6: Dabat in 2023 cropping season. The blue numbers connected to the polygon represent the genotypes, whereas the green numbers represent the locations/test environments.

in one are likely to perform poorly in the other, and vice versa [61,66]. In the present study, as shown in (Fig 5), the environments Simada, and Bichena are positively correlated, as indicated by an angle their vectors less than 90°. On the other hand, the angle between Dabat and simada greater than 90°, indicating a negative correlationbetween these two environments.

### 3.6. Stability of durum wheat genotypes

Graphical representation of genotypes' mean grain yield and stability is shown using the average environment coordination (AEC) method (Fig 6). Combining grain yield with genotype stability performance help identify genotypes that yield the highest and most stable results [78]. The genotype with the highest mean performance and greatest stability across all test environments is considered ideal [79,80]. According to the AEC view in the comparison biplot, a desirable genotype are those located close to the ideal genotype, which is typically positioned at the center of the concentric circles or at the tip of the arrow indicating the ideal point. In the AEC (Average Environment Coordination) system, the mean performance of genotypes is represented by an arrow along the AEC X-axis (PC1), which passes through the origin of the biplot and indicates the average environment axis, reflecting the overall performance of genotypes across environments.This graphical representation helps visually distinguish high-performing, stable genotypes from those with variable or suboptimal performance.

On the positive side of the AEC abscissa indicates that genotypes closer to the origin have a higher mean grain yield, while genotypes farther away on the negative side have a lower mean grain yield. Additionally, a genotype's projection becomes less stable the farther it is from the origin in absolute terms. This line indicates a higher mean yield across the environments. Hence, in the present study, G2 gave the highest mean yield, followed by G1, and G6. The performance of the remaining genotype groups (G3 and G5) was below the grand mean yield (Fig 6). An ideal genotype is completely stable across a wide range of environments and has the highest average performance of any genotype [81].

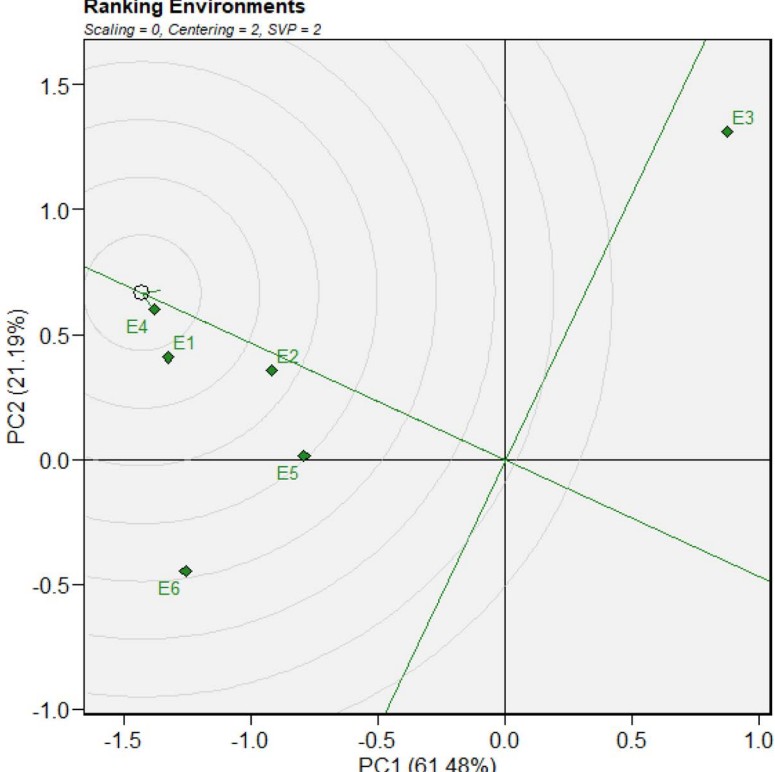

**Fig 5. Ranking of environments relative to the ideal environments.** Where: E1: Simada, E2: Bichena, E3: Dabat 2022 cropping season and E4: Simada, E5: Bichena, E6: Dabat in 2023 cropping season.

Higher GEI interaction and lower stability are indicated by genotype that lie further from the origin along the AEC ordinate (y-axis), in either direction [82]. G4 (Tesfaye) and G2 (Don Matteo) showed high yield and stability. G8 (Mangudo) was the most stable but moderate in yield. Conversely, G3 (Alemtena) and G6 (Rigeat) showed instability and were adapted to specific environments. Although G3 (Alemtena) produced the highest mean yield in some environments, its large deviations along the AEC ordinate indicated low stability (Fig 6). Among environments, E2 (Simada 2022), E4 (Bichena 2023), and E1 (Bichena 2022) were located closer to the AEC abscissa, indicating representativeness, while E3 (Dabat 2022) showed the largest deviation, suggesting high discriminating ability but low representativeness.

An ideal genotype for a specific environment has the highest mean yield and responds best in that particular environment while being less stable in the other environments, thus necessitating recommendations for the specific environment [83]. Stability is reported to have lower heritability than mean performance [69,84]. It is important to note that stability pertains to the genotype's relative performance and gains significance only when linked to the mean [85]. In summary, environmental factors (e.g., temperature, rainfall, soil quality) and genetic factors (e.g., drought resistance, nutrient efficiency) jointly influence yield-related parameters such as biomass, spike length, and the number of tillers. Wheat varieties with favorable genetic factors tend to perform better under various environmental conditions, resulting in improved overall yield [13,86,87]. Genotype stability was further evaluated using different stability parameters. The figure clearly identifies G4 as the ideal genotype and E4 as the ideal environment. G1 and G2 also appear as desirable genotypes, while E1 and E2 emerge as desirable environments, suggesting they offer favorable conditions for good performance and stability.

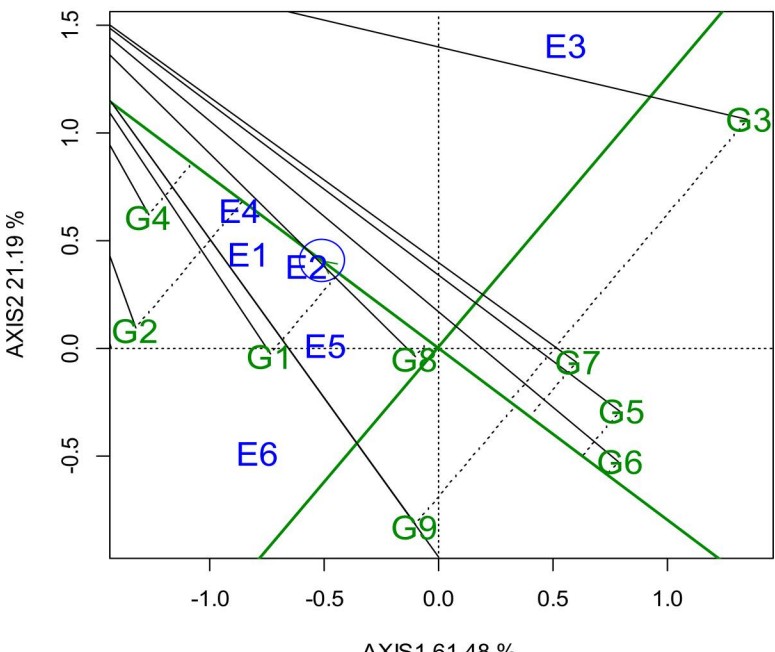

**Fig 6. Which genotype was more stable across the tested environments. ?** Where: G1: Fetan, G2: Don Matteo, G3: Alemtena, G4: Tesfaye, G5: Wehabit, G6: Rigeat, G7: Utuba, G8:Mangudo, and G9: Toltu genotype (E1: Simada, E2: Bichena, E3: Dabat 2022 cropping season and E4: Simada, E5: Bichena, E6: Dabat in 2023 cropping season. The blue numbers connected to the polygon represent the genotypes, whereas the green numbers represent the locations/test environments.

### 3.7. Stability parameter

**3.7.1. The regression coefficient.** The performance of a genotype in an environment is determined by its mean performance, environmental responsiveness as a linear function, and its deviation from regression. Eberhart and Russell suggested using a linear regression coefficient and the variance of the regression deviations to evaluate crop responses to environmental changes. The average stability is indicated by a regression coefficient ($b_i$) that approaches one and a deviation from regression ($S^2di$) of zero. In this model, regression scores above one indicate varieties that are more sensitive to changes in the environment (below-average stability) and are specifically adaptable to high-yielding environments [88]. Conversely, regression coefficients below one enhance the specificity of adaptability to low-yielding environments by demonstrating greater resistance to environmental change (above-average stability).

The present results indicated that the linear regression of the average grain yield of individual genotype on the average yield of all varieties in each environment produced regression coefficients ($b_i$ values) ranging from −0.012 to 1.73. This variation in regression coefficients reflects the differing responses of the varieties to environmental changes (Table 7). A mean regression coefficient ($b_i$) close to one, low deviation form regression, and grain yields higher than the grand mean indicate that the aforementioned varieties Fetan, Tesfaye and Utuba are well adapted to all environments, indicating broader adaptation across different environments. In contrast Toltu is poorly adapted to all environments. It is recommended that these varieties be cultivated under unfavorable conditions, as they demonstrate resilience to environmental variations. Similar finding have been reported by other researchers [89].

**Table 7. The various models of stability used to partition the GEI interaction for grain yield in the test of durum wheat genotypes and their ranking.**

| Genotypes | Mean | Rank | Bi | s²di | σ² | Wi | Pi | ASU | YSI | Rank | s1 | s2 | ASI | ASV |
|---|---|---|---|---|---|---|---|---|---|---|---|---|---|---|
| 1 | 3.3 | 3 | 0.84 | 0.14 | 0.12 | 97.70 | 0.11 | 7 | 10 | 4 | 0.27 | 7.77 | 0.28 | 1.38 |
| 2 | 3.51 | 2 | 1.41 | 0.21 | 0.22 | 102.00 | 0.00 | 8 | 10 | 3 | 0.33 | 10.70 | 0.40 | 2.00 |
| 3 | 3.02 | 6 | −0.012 | 0.62 | 0.74 | 80.30 | 0.80 | 9 | 15 | 6 | 0.20 | 8.67 | 0.76 | 3.78 |
| 4 | 3.69 | 1 | 0.98 | 0.04 | 0.03 | 114.00 | 0.00 | 5 | 6 | 1 | 0.07 | 1.47 | 0.18 | 0.87 |
| 5 | 2.75 | 9 | 0.35 | 0.00 | 0.03 | 84.70 | 0.79 | 2 | 11 | 9 | 0.40 | 4.97 | 0.11 | 0.54 |
| 6 | 2.78 | 8 | 1.32 | 0.06 | 0.05 | 83.30 | 0.66 | 3 | 11 | 8 | 0.33 | 4.57 | 0.11 | 0.56 |
| 7 | 2.95 | 7 | 0.73 | 0.04 | 0.03 | 90.00 | 0.56 | 4 | 11 | 7 | 0.13 | 3.50 | 0.17 | 0.87 |
| 8 | 3.12 | 4 | 1.43 | 0.14 | 0.20 | 91.30 | 0.27 | 1 | 5 | 2 | 0.27 | 8.80 | 0.10 | 0.48 |
| 9 | 3.07 | 5 | 1.73 | 0.22 | 0.27 | 87.40 | 0.45 | 6 | 11 | 5 | 0.00 | 14.60 | 0.27 | 1.33 |

$S^1$ = Nassar and Huens's (1987) Mean of the absolute rank differences of a genotype over the n environments, $S^2$ = Nassar and Huens's (1987) variance among the ranks over the k environments, **Wi** = Wriecke's ecovalence, **Pi** = Cultivar superiority, $\sigma^2$ = Shukla's (1972) stability variance, **bi** = Eberhart and Russell (1966) stability value of regression coefficient and $S^2di$ = Eberhart and Russell (1966) stability deviation value from regression, ASI = AMMI stability index, and ASV = AMMI stability value.

### 3.8. Limitation of the study

While the findings provide valuable insights into the performance and stability of durum wheat genotypes, it is important to acknowledge certain limitations. The study is based on data collected over only two years and across six environments, which may not fully capture the extensive environmental variability affecting durum wheat production. Additionally, although the statistical methods used are robust, they carry inherent assumptions and limitations that could influence the interpretation of genotype-by-environment interactions.Therefore, recommendations for commercial cultivation should be made with caution. Conducting further multi-year and multi-locati on trials would enhance the reliability and generalizability of these results.

### 4. Conclusions and recommendation

Analysis of GEI interaction in this study highlighted the importance of evaluating durum wheat genotypes across multiple environments to determine their yield stability and adaptability. The combined analysis of variance over the years showed significant difference among genotypes, environments, and their interaction, that both genetic and environmental factors significantly influenced grain yield and other agronomic traits. Genotype Tesfaye, followed by Don Matteo, consistently exhibited superior phenotypic performance across all traits,with grain yields of 3.74 and 3.62 tha⁻¹, respectively, making them the top-yielding genotypes. AMMI and GGE biplot analyses effectively visualized the GEI patterns, grouping the six test environments and nine genotypes into two mega-environments and six genotype clusters. Simada and Bichena emerged as both highly discriminating and representative environments, making them ideal sites for evaluating genotype performance. Grain yield analysis showed that Tesfaye (G4) and Don Matteo (G2) consistently achieved the highest yields and broad adaptability, whereas Wehabit (G5) and Rigeat (G6) performed poorly, reflecting limited stability. Based on these results, it is recommended to cultivate Tesfaye and Don Matteo in the Amhara region and other similar durum wheat-growing areas, given their high yield and moderate to high stability. Additionally, Simada and Bichena should be prioritized as core testing sites in future multi-environment trials due to their strong discriminative and representative capacities. Continued exploration of GEI is essential to refine varietal recommendations and ensure optimal wheat production in specific agro-ecological zones.

### Supporting information

**S1 Table. All Stability Analysis of Genotypes Across Environments.** This table presents the stability indices of genotypes, showing their performance variation and adaptability across different environments.
(XLSX)

**S2 Table. Durum wheat row data.** This table presents the raw grain yield data of durum wheat genotypes collected from different environments before stability analysis.
(XLSX)

**S3 Table. Mean yield and AMMI IPCA scores durum wheat genotypes across test environments.** Mean grain yield and AMMI Interaction Principal Component Axis (IPCA) scores of durum wheat genotypes across test environments.
(CSV)

## Author contributions

**Conceptualization:** Alemnesh Eskezia.

**Data curation:** Alemnesh Eskezia, Alamir Ayenew.

**Formal analysis:** Alemnesh Eskezia, Yohannes Azene, Kelemu Nakachew.

**Funding acquisition:** Alemnesh Eskezia.

**Investigation:** Alamir Ayenew, Kelemu Nakachew.

**Methodology:** Alemnesh Eskezia, Alamir Ayenew.

**Software:** Alemnesh Eskezia, Yohannes Azene.

**Supervision:** Alemnesh Eskezia, Yohannes Azene, Mulugeta Aytenew.

**Validation:** Alemnesh Eskezia, Alamir Ayenew, Mulugeta Aytenew.

**Visualization:** Mulugeta Aytenew, Kelemu Nakachew.

**Writing – original draft:** Alemnesh Eskezia.

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
