## [Decision Letter · Decision Letter 0]

22 Jul 2025

Dear Dr. Eskezia,

Thank you for submitting your manuscript to PLOS ONE. After careful consideration, we feel that it has merit but does not fully meet PLOS ONE’s publication criteria as it currently stands. Therefore, we invite you to submit a revised version of the manuscript that addresses the points raised during the review process.

We look forward to receiving your revised manuscript.

Kind regards,

Mehdi Rahimi, Ph.D.

Academic Editor

PLOS ONE

Journal Requirements:

3. In the online submission form, you indicated that [Datasets generated in this study are available from the corresponding author upon reasonable request.].

In the figure caption of the copyrighted figure, please include the following text: “Reprinted from [ref] under a CC BY license, with permission from [name of publisher], original copyright [original copyright year].

Reviewers' comments:

Reviewer's Responses to Questions

**Comments to the Author**

1. Is the manuscript technically sound, and do the data support the conclusions?

Reviewer #1: Yes

Reviewer #2: Partly

2. Has the statistical analysis been performed appropriately and rigorously?

Reviewer #1: No

Reviewer #2: Yes

3. Have the authors made all data underlying the findings in their manuscript fully available?

Reviewer #1: Yes

Reviewer #2: No

4. Is the manuscript presented in an intelligible fashion and written in standard English?

Reviewer #1: Yes

Reviewer #2: Yes

Reviewer #1: This paper has good data but the biggest problems are here;

1) Which factors are random or fixed, especially the locations.

2) Combining year by location to create an environment will then NOT permit most of the analysis done to be feasible, like representative environment, discriminating environment and which won where. Only stable varieties can be identified but you cannot say the best genotype for 2022/2023 in Simada is genotype XX, No. Please refer to the following paper Yan and Tinker, 2005; and yan and Kang, 2002 when the introduced the GGE biplot concept.

3) The model diagnostic is key when you do AMMI, read Gauch 2013, A simple protocol for AMMI.

Please read those articles well and re-do the paper.

Reviewer #2: Please see the attachmeng file 'review report ...' I am unable to write the full report here because of logistic problems occurred temporarily. I am submitting it using my smartphone & there is poor connection

**Do you want your identity to be public for this peer review?** For information about this choice, including consent withdrawal, please see our Privacy Policy

Reviewer #1: **Yes:** Edmore Gasura

Reviewer #2: No

---

## [Author Response · Author response to Decision Letter 1]

18 Sep 2025

Manuscript. Number: PONE-D-27-30070

Journal: Plos one

Title: Genotype by Environment Interaction and Yield Stability Analysis of Durum Wheat (Triticum turgidum var. durum) Varieties Using AMMI and GGE biplot Analysis in Amahara Region, Northwest Ethiopia

Dear Editor and Reviewers

We would like to thank both the editor and the reviewers for your contribution to this manuscript by filtering out the technical and editorial errors in the scientific research writing that could increase the reliability and scientific exchange of the research output. We also admire your effort to see the details of our manuscript taking your golden time and we value all the comments. Further, we would be very glad to publish our manuscript in the Plos one journal if accepted. We believe that we have addressed all comments as suggested in the text and point-by-point response. Hence, kindly consider our revisions accordingly.

Response to the comments

Title section reviewer comment.

Title: needs minor modification and I suggest it to be;

“Genotype by Environment Interaction and Yield Stability Analysis of Durum Wheat (Triticum turgidum var. durum) Varieties Using AMMI and GGE biplot Analysis in Amahara state or Region, Northwest Ethiopia”.

Response_ Thank you for your valuable comments. We have modified our manuscript title based on the reviewer’s feedback and suggestions.

Genotype by Environment Interaction and Yield Stability Analysis of Durum Wheat (Triticum turgidum var. durum) Varieties Using AMMI and GGE biplot Analysis in Amahara Region, Northwest Ethiopia

Abstract section reviewer comment.

Abstract: this section needs major modification;

1. Abstract poorly composed though the study has good results.

Response_ We have updated the section by incorporating the main results into the abstract.

2. It should be written again because some data presented there has mistakes

Response: We have rewritten and updated the entire abstract based on the study results.

3. Some important result information not presented

Response_ We have updated it

4. There are lots of grammatical & typographical mistakes

Response_ All grammatical and typographical errors have been corrected as per the reviewer’s comment.

5. I tried to correct some of them using the MSWord track review method in the attached file

Response_Thank you for your valuable comments. We have reviewed them and made the necessary corrections.

6. The AMMI analysis Anova output was reported as the result of combined anova result

Response_ We have corrected the AMMI analysis ANOVA output to accurately present it separately from the combined ANOVA results.

Introduction section reviewer comment.

1. It has good information but presented poorly, it is highly disorganized, paragraphs should be reshuffled.

Response_ the necessary corrections have been made, and the paragraphs have been reorganized to improve clarity and structure.

2. In text citations didn’t follow the journal’s guideline

Response_ In-text citations have been revised in accordance with the journal’s guidelines.

3. Lots of sentences not clearly written. Several of them need to be rewritten

Response_ We have rewritten unclear sentences to improve clarity and readability

4. Some topics like AMMI & GGE described repeatedly about their use and assumptions in detail, but no previous literatures on durum wheat presented that show their practical applications

Response_ We have revised the sections on AMMI and GGE to reduce repetition and included relevant previous literature on their applications in related crops.

5. There is no statement or justifications why the authors selected the AMMI & GGE biplot models

Response_ We have added a clear justification for selecting the AMMI and GGE biplot models, highlighting their suitability for analyzing genotype by environment interactions in our study

6. Problem statement or the need to do this study not described sufficiently

Response_ The problem statement has been strengthened to provide a more comprehensive justification for conducting this study.

7. Objectives of the research not stated clearly

Response _We have revised the objectives section to clearly and explicitly state the research goals

Materials & methods section reviewer comment.

1. the sub-title 2.1 should be modified; here you are giving the detail account of the study (experimental) area

Response _ we update it

2. Table 1.Geographical description of the study area; It would be more informative if you can incorporate air temperature, and rainfall data of each year for the specific locations for the specific years. Specific geographic coordinates (including altitude) of the trial sites & if possible soil physic-chemical properties.

Response _ We have included air temperature and rainfall data for each year and the specific geographic coordinates, including altitude, of the trial sites. However, soil physicochemical properties were not available and thus could not be included

3. 2.2. Experimental material and design; first 4-5 line needs to be rewritten

Response-We have rewritten the first 4–5 lines of section 2.2 to improve clarity and coherence

4. Table 3; Can you include their pedigree for each variety? You better include their maturity period, yielding ability, areas of adaptation (recommendation), level of disease resistance, etc.

Response- We have included the pedigree, maturity period, yielding ability, recommended areas of adaptation, and disease resistance levels for each variety in Table 3 as suggested.

5. 2.3. Data Collected; you have not described how you collected the data. At least you can insert citations of a published material that explains how you collected the data. So that readers get knowledge of how to do similar experiments using same or different materials.

Response- We have included a detailed description of the data collection methods along with relevant citations to guide readers on how to conduct similar experiments.

6. GGE biplot analysis- Which method or formula did you use? You need to show the formula you have used to compute the analysis

Response- The manuscript now includes a detailed description of the GGE biplot analysis method along with the corresponding formula, as requested

Results and discussions section reviewer comment

• Same problem was observed in the in text citation as above

1. 3.2. Performance of genotypes - What about the performance of these durum wheat genotypes in previous studies by other researchers? Please cite previous works that either agree or contradict with your findings to make your discussion complete

Response- Relevant literature discussing the performance of durum wheat genotypes in previous studies has been cited to contextualize and strengthen the discussion of our results.

2. What is this MSE? Is it the std of the mean values?

Response-Its MSE (Mean Square Error) is not the standard deviation.

3. 3.3. Genotype performance at each tested location- this sub-topic also lacks supporting literatures for the findings presented similar to sub-topic 3.2

Response- We have added relevant supporting literature to the sub-topic on genotype performance at each tested location.

4. 3.4. Additive Main Effects and Multiplicative Interaction (AMMI) Analysis- One table is missing from AMMI analysis; the significant IPCAs (usually 1&2) give vector values for each genotype that indicates how much each genotype contributes to the GEI. Absolute value of the IPCA score determines its stability. Low value close to zero indicates the genotype being stable. You didn’t indicate which genotypes are stable using the AMMI analysis either using the table or the AMMI biplot 1 graph.

Response - We have included the AMMI analysis results by adding Table 7, which presents YSI, ASV, IPCA1, and IPCA2 values. Based on these, we have identified and indicated the stable genotypes as per the reviewer’s suggestion.

5. Please refer the following publication for better understanding of the results and interpretation of AMMI analysis; Kassie MM, Abebe TD, Desta EA, Tadesse W. Multi-Environment Evaluation of Bread Wheat Genotypes for Yield Stability in Ethiopia Using AMMI and GGE-Biplot Analyses. Crop Breed Genet Genom. 2025;7(2):e250004.

Response - Thank you for the helpful reference. We have reviewed the suggested publication by Kassie et al. (2025) and incorporated relevant insights to enhance the interpretation and discussion of the AMMI analysis results in our manuscript.

6. It appears that there is error in the table numbering, check it, see the comment in the manuscript file

Response - We have carefully reviewed the manuscript and corrected the table numbering errors as pointed out.

7. On page 17, the values you presented from table 7 about the proportions of the sum of squares of the AMMI anova seems to be incorrect. In addition, another table that lists IPCA scores for each genotype & even environments is missing. The IPCA scores help in identifying the relative contribution of genotypes for the GEI determines which genotype is stable/unstable. Please read the paper Kassie, et al (2025) and some recently published papers in durum wheat by Liyew et al. (2025) too.

Response -We have corrected the values from Table 7 regarding the sum of squares proportions in the AMMI ANOVA and added the IPCA scores for each genotype and environment as subcomponents within Table 7. This addition facilitates identification of genotype stability and contribution to GEI. We also reviewed and incorporated insights from Kassie et al. (2025) and Liyew et al. (2025) to strengthen our analysis.

8. Figure 3, the mega environment grouping not clearly displayed

Response – We have updated Figure 3 to clearly display the mega-environment groupings, making them easier to identify

9. Figure 4, concentric circles not visible

Response – We have enhanced Figure 4 to improve the visibility of the concentric circles as per your comment

10. Page 26, there is unfinished sentence; either correct it or delete it.

Response – he unfinished sentence on page 26 has been corrected to ensure clarity and completeness

11. 3.7 stability parameter; here, you have produced a beautiful table (the last table) for the stability analysis, but it was not exploited fully. You explained only one of the stability models that favors high grain yield. Please use the most recent stability models too like ASV, YSI

Response –Thank you for the valuable suggestion. We have expanded the stability analysis section to include and discuss additional recent stability models such as ASV and YSI, providing a more comprehensive interpretation of the results

12. References; Please use the journal’s guideline for in text citations & style of referencing

Response – We have carefully revised the in-text citations and reference list to fully comply with the journal’s guidelines

---

## [Decision Letter · Decision Letter 1]

8 Oct 2025

Dear Dr. Eskezia,

Thank you for submitting your manuscript to PLOS ONE. After careful consideration, we feel that it has merit but does not fully meet PLOS ONE’s publication criteria as it currently stands. Therefore, we invite you to submit a revised version of the manuscript that addresses the points raised during the review process.

We look forward to receiving your revised manuscript.

Kind regards,

Mehdi Rahimi, Ph.D.

Academic Editor

PLOS ONE

Journal Requirements:

Reviewers' comments:

Reviewer's Responses to Questions

**Comments to the Author**

Reviewer #2: All comments have been addressed

2. Is the manuscript technically sound, and do the data support the conclusions?

Reviewer #2: Yes

3. Has the statistical analysis been performed appropriately and rigorously?

Reviewer #2: Yes

4. Have the authors made all data underlying the findings in their manuscript fully available?

Reviewer #2: No

5. Is the manuscript presented in an intelligible fashion and written in standard English?

Reviewer #2: No

Reviewer #2: There are comments and suggestion made on the manuscript that needs your attention for further improvement. Please see the attached pdf file of your manuscript for modification.

**Do you want your identity to be public for this peer review?** For information about this choice, including consent withdrawal, please see our Privacy Policy

Reviewer #2: No

---

## [Author Response · Author response to Decision Letter 2]

23 Dec 2025

Manuscript. Number: PONE-D-27-30070 R2

Journal: PLOS ONE

Title: Genotype by Environment Interaction and Yield Stability Analysis of Durum Wheat (Triticum turgidum var. durum) Varieties Using AMMI and GGE biplot Analysis in Amahara Region, Northwest Ethiopia

Dear Editor and Reviewers

We would like to thank both the editor and the reviewers for your contribution to this manuscript by filtering out the technical and editorial errors in the scientific research writing that could increase the reliability and scientific exchange of the research output. We also admire your effort to see the details of our manuscript taking your golden time and we value all the comments. Further, we would be very glad to publish our manuscript in the Plos one journal if accepted. We believe that we have addressed all comments as suggested in the text and point-by-point response. Hence, kindly consider our revisions accordingly.  

Response to the General comment

1. The manuscript requires thorough English language editing to improve grammar, clarity, and readability.

Response- We have carefully edited and reviewed the entire document to improve its English language quality and grammar throughout.

2. We have carefully addressed each reviewer’s comment point by point throughout the main document. All corrections and revisions have been made according to the reviewers’ suggestions and recommendations, and the changes are highlighted in color for easy reference.

Response - Thank you for this valuable comment. We have carefully reviewed all references cited in the manuscript. Retracted or outdate articles have been removed and replaced with recent and relevant publications. Additionally, all remaining references have been verified for accuracy, completeness, and citation consistency. The updated reference list now includes only valid and current sources that directly support the findings of our study.

Based on the reviewer’s comment, here are the specific actions you must take:

1. Review all cited references — check that each one is relevant to your study and supports the statements in your manuscript.

2. Check for retracted papers — if any of your cited articles have been retracted, either:

o Remove them and replace them with current, valid references, or

o If you must cite them (rarely), clearly state that they are retracted and explain why they are still relevant.

3. Verify accuracy — ensure all reference details (authors, year, title, journal, volume, pages, DOI) are correct and properly formatted.

4. Add missing references — include any relevant studies suggested by reviewers if they strengthen your work.

5. Mention these updates in your rebuttal letter — briefly note that the reference list has been reviewed, corrected, and updated as per the reviewer’s suggestion.

Response- Thank you for the valuable feedback. We have carefully reviewed and updated the reference list as per the reviewer’s suggestions. All cited works have been checked for relevance, accuracy, and retraction status.

---

## [Editor Report · Decision Letter 2]

22 Jan 2026

Genotype by Environment Interaction and Yield Stability Analysis of Durum Wheat ( Triticum turgidum var. durum) Varieties Using AMMI and GGE biplot Analysis in Amahara Region, Northwest Ethiopia

PONE-D-25-30070R2

Dear Dr. Eskezia,

We’re pleased to inform you that your manuscript has been judged scientifically suitable for publication and will be formally accepted for publication once it meets all outstanding technical requirements.

Kind regards,

Mehdi Rahimi, Ph.D.

Academic Editor

PLOS One
---

## [Editor Report · Acceptance letter]

PONE-D-25-30070R2

PLOS One

Dear Dr. Eskezia,

I'm pleased to inform you that your manuscript has been deemed suitable for publication in PLOS One. Congratulations! Your manuscript is now being handed over to our production team.

Kind regards,

on behalf of

Associate Prof. Mehdi Rahimi

Academic Editor

PLOS One